# Potentiality and Inflammatory Marker Expression Are Maintained in Dental Pulp Cell Cultures from Carious Teeth

**DOI:** 10.3390/ijms23169425

**Published:** 2022-08-20

**Authors:** Shelly Arora, Paul R. Cooper, Lara T. Friedlander, Benedict Seo, Shakila B. Rizwan, Alison M. Rich, Haizal Mohd Hussaini

**Affiliations:** 1Sir John Walsh Research Institute, Faculty of Dentistry, University of Otago, P.O. Box 56, Dunedin 9054, New Zealand; 2School of Pharmacy, University of Otago, P.O. Box 56, Dunedin 9054, New Zealand; 3Faculty of Dental Medicine, Airlangga University, Surabaya 60132, Jawa Timur, Indonesia

**Keywords:** dental pulp stem cells, immune response, mesenchymal stem cells, tooth pulp

## Abstract

Objectives: This investigation aimed to isolate and culture human dental pulp cells from carious teeth (cHDPCs) and compare their growth characteristics, colony-forming efficiency, mineralization potential and gene expression of Toll-like receptors (TLR)-2, TLR-4, TLR-9, tumour necrosis factor (TNF)-α, interleukin (IL)-1β, IL-6, IL-8, IL-17A, 1L-17R, IL-23A, nuclear factor-kappa B (NF-κB), mitogen-activated protein kinase (MAPK1), dentin matrix protein (DMP)-1, dentin sialophospho protein (DSPP), sex determining region Y-box 2 (SOX2) and marker of proliferation Ki-67 (MKi67) with cells isolated from healthy or non-carious teeth (ncHDPCs). Methods: Pulp tissues were obtained from both healthy and carious teeth (n = 5, each) to generate primary cell lines using the explant culture technique. Cell cultures studies were undertaken by generating growth curves, a colony forming unit and a mineralization assay analysis. The expression of vimentin was assessed using immunocytochemistry (ICC), and the gene expression of above-mentioned genes was determined using quantitative real-time reverse-transcription polymerase chain reaction. Results: ncHDPCs and cHDPCs were successfully isolated and cultured from healthy and inflamed human dental pulp tissue. At passage 4, both HDPC types demonstrated a typical spindle morphology with positive vimentin expression. No statistical difference was observed between ncHDPCs and cHDPCs in their growth characteristics or ability to differentiate into a mineralizing phenotype. ncHDPCs showed a statistically significant higher colony forming efficiency than cHDPCs. The gene expression levels of TLR-2, TLR-4, TLR-9, TNF-α, IL-6, IL-8, IL-17R, IL-23A, NF-κB, MAPK1, DMP1, DSPP and SOX2 were significantly higher in cHDPCs compared with ncHDPC cultures. Conclusion: cHDPCs retain their differentiation potential and inflammatory phenotype in vitro. The inflamed tooth pulp contains viable stem/progenitor cell populations which have the potential for expansion, proliferation and differentiation into a mineralizing lineage, similar to cells obtained from healthy pulp tissue. These findings have positive implications for regenerative endodontic procedures.

## 1. Introduction

Primary cells harvested from dental pulp tissue are termed human dental pulp cells (HDPCs) and this heterogeneous population contains several cell types, including fibroblasts, undifferentiated mesenchymal/progenitor cells, immune cells, nerve cells, perivascular and endothelial cells [1,2]. HDPC cultures are considered to represent the pulp tissue itself as they can resemble its complexity when utilized in a laboratory setting [3]. As a result, in vitro HDPC cultures have been used to predict the behaviour of the pulp in vivo, such as in response to dental materials, disease, or other changes in the local tissue environment [4].

HDPCs contain resident stem cells termed dental pulp stem cells (DPSCs). DPSCs are multipotent mesenchymal cells and have the potential for self-renewal and multiple differentiation [5]. In vitro DPSCs can differentiate into a range of cell types, including cells with a mineralizing phenotype, such as odontoblasts-like cells and osteoblasts [6] and in vivo they can form dentine-pulp-like complexes [7]. The potential of DPSCs makes them a promising tool for cell-based therapies, including regenerative endodontic procedures (REPs). REPs represent an emerging clinical field and are defined as: ‘*biologically based procedures designed to replace damaged tooth structures, including dentine, root structures, as well as cells of the pulp–dentine complex*’ [8].

Dental caries is the most common cause of pulp inflammation and results from microbial invasion and their pro-inflammatory by-products. [9]. The inflamed pulp is characterised by increased levels of cytokines, including tumour necrosis factor (TNF)-α, interleukin (IL)-1α, IL-1β, IL-4, IL-6, IL-8, IL-17 and IL-23, which recruit and drive the complex cellular immune response [10,11,12,13,14,15]. This local inflammatory microenvironment may subsequently alter the behaviour and phenotype of HDPCs [16]. HDPCs derived from healthy or non-carious teeth (ncHDPCs) have been extensively studied; however, there is only limited literature describing the isolation and characterization of HDPCs derived from carious teeth (cHDPCs), the cellular and molecular differences between ncHDPCs and cHDPCs, or whether cHDPCs exhibit differentiation potential [17,18,19,20]. Their application for cell-based therapies, including REPs, requires a better understanding. Nevertheless, previous studies on cHDPCs have shown that cHDPCs have higher proliferation and mineralization potential than ncHDPCs [17,18]. Therefore, this study focuses on the isolation and expansion of HDPCS from healthy and carious teeth, their characterization and comparison in terms of growth, colony forming capacity, differentiation potential and gene expression.

## 2. Results

### 2.1. Establishment and Characterization of Primary Cell Cultures

HDPCs were successfully isolated and cultured from healthy and carious teeth using the dental pulp explant method. Five unerupted healthy maxillary third molars were collected from five patients with an age range of 19–25 years; three were male and two were female. Five carious teeth (maxillary and mandibular molars) were collected from five patients with an age range of 29–35 years; four were females and one male.

Both ncHDPCs and cHDPCs on average required a similar number of days (16 days) to achieve initial confluence (P0). There was no microbial contamination detected during cell culture in either group. No morphological differences were apparent between ncHDPCs and cHDPCs and both were spindle-shaped and fibroblast-like (Figure 1(Ai,Aii)). Positive cytoplasmic staining with anti-vimentin antibody was detected in both the ncHDPCs and cHDPCs, indicating their mesenchymal origin (Figure 1(Bi,Bii)).

At day one, a lag period of growth was observed in both cell populations (ncHDPCs and cHDPCs) as shown by the decline in cell number from the original seeding density. In general, both cell groups entered the log phase from day two and continued to proliferate up to day seven (Figure 2A). Comparison between ncHDPCs and cHDPCs was conducted at each time point; however, there was no significant difference in the proliferation between the two cell populations (*p* = 0.8889).

### 2.2. Differentiation Potential of Human Dental Pulp Cell Cultures

The crystal violet staining of HDPC cultures derived from healthy and carious teeth demonstrated the ability of both cell populations to form colonies (Figure 2(Bi,Bii)). ncHDPCs showed an average of 152.67 ± 26 colonies, whereas cHDPCs showed an average of 73.53 ± 32 colonies formed. This difference was statistically significant (**** *p* < 0.0001) (Figure 2(Biii)).

**Figure 2 ijms-23-09425-f002:**
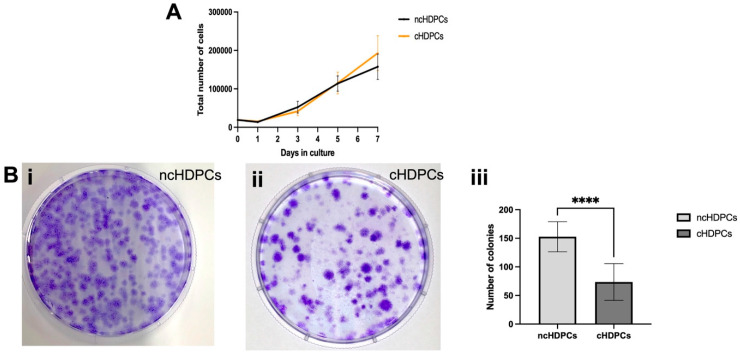
(**A**) Growth curves for ncHDPCs and cHDPCs (n = 5 each) over 7 days of culture. No significant differences were observed between the two cell populations (*p* = 0.8889). Error bars indicate mean ± SD. (**B**) Colony forming unit (CFU) efficiency of ncHDPCs and cHDPC. Representative images of crystal violet-stained colonies at 14 days (**i**,**ii**) are shown. (**iii**) Quantification of the average number of CFUs formed for ncHDPCs and cHDPCs (n = 5 each). A statistically significant difference was found in the colony forming efficiency between ncHDPCs and cHDPCs (**** *p* < 0.0001).

Microscopic examination of Alizarin red S-stained cultures showed the presence of mineralized nodules (red colour) indicative of dental pulp cell differentiation (Figure 3(Ai,Aii)). ncHDPCs and cHDPCs cultured in a control medium (DMEM containing 10% FBS without mineralizing supplements) did not form mineralized deposits (Figure 3(Bi,Bii)). The mineralization matrix quantification assay showed no statistically significant difference between ncHDPC and cHDPC cultures under mineralizing conditions for 21 days with respect to the amount of mineralized matrix produced (*p* = 0.18) (Figure 3C).

### 2.3. Gene Expression Analysis in Cultures of Non-Carious and Carious Human Dental Pulp Cells

The mRNA expression level of TLR4 (fold change = 2.32; *p* < 0.0001) was significantly upregulated while TLR2 (fold change = 1.40; *p* = 0.04) and TLR9 (fold change = 1.24; *p* = 0.01) were also detected towards higher levels in the cHDPCs compared with the ncHDPC samples. Among the cytokines assayed, TNF-α showed the highest increase in mRNA expression, a 12-fold increase (*p* < 0.0001) in cHDPCs. In addition, other cytokines, including IL-6 (fold change = 5.91; *p* < 0.0001), IL-8 (fold change = 6.35; *p* < 0.0001), IL-17R (fold change = 2.00; *p* < 0.0001) and IL-23A (fold change = 1.90; *p* < 0.0001), were also significantly more abundantly expressed in the cHDPC cultures compared with the ncHDPC cultures. There was no significant difference in the mRNA expression of IL-1β (fold change = 1.77; *p* ≥ 0.99). The IL-17A transcript was not detected in ncHDPCs and cHDPCs. Consistent with the upregulated cytokine data, the transcript levels of the regulatory markers NF-κB (fold change = 1.27; *p* = 0.01) and MAPK1 (fold change = 1.21; *p* = 0.01) were found to be higher in cHDPCs as compared with ncHDPCs. The mRNA levels of DMP1 (fold change = 6.94; *p* < 0.0001) and DSPP (fold change = 19.58; *p* < 0.0001) were also more abundantly expressed in cHDPC cultures. SOX2 gene expression (fold change = 12.10; *p* < 0.0001) level was also significantly higher in the cHDPCs compared with the ncHDPCs. The expression level of MKi67 (fold change = 1.37; *p* = 0.06) was not significantly different in its expression between ncHDPCs and cHDPCs (Figure 4).

## 3. Discussion

The results of this study demonstrate that HDPCs from both healthy and carious mature teeth can be successfully isolated and cultured. Whilst this is in agreement with some previous studies [17,18,19,20], others have reported limited success in isolating HDPCs from diseased pulpal tissue. This outcome was previously attributed to there being only a relatively small amount of viable pulp tissue obtained from carious teeth and hence this corresponded with a lower likelihood of establishing successful cell cultures [17,21]. The present study did not experience any issues in establishing primary pulp cell cultures and a possible explanation for this success was that pulp tissue was extracted from carious molars, which have a relatively larger pulp volume compared with other tooth types, and the the pre-operative radiographic staging of carious lesions ensured the pulp space was sufficiently visible [22]. Further, this also clearly indicates that the teeth included in this study have a vital pulp. Notably, no differences were found in the morphology of ncHDPCs and cHDPCs with both displaying spindle-shaped fibroblast morphology (Figure 1A,B) and vimentin positivity, which also corroborates previous findings [18,20,23,24,25]

Notably, our study identified that significantly more cells colonies from ncHDPC populations were generated in comparison with cHDPCs (Figure 2A) which is consistent with some previous findings [26,27]. Indeed, it has been previously proposed that cHDPCs isolated from the inflamed tooth pulp displayed reduced colony forming capacity due to the presence of the inflammatory micro-environment as this potentially inhibits cell division processes [26,27,28]. In contrast, others have reported a higher clonogenic potential in cHDPCs compared with ncHDPCs and that this increased capacity of cHDPCs was associated with the retention of the inflammatory phenotype in vitro [17,18]. Interestingly, further analysis of our data indicated that in subsequent passages both ncHDPCs and cHDPCs showed comparable growth rates (Figure 2B). These data were consistent with our MKi67 proliferation-associated transcript data, which demonstrated similar levels of expression and are in agreement with those previously reported by Pereira et al. [21]. This suggests that both groups of cells maintained their proliferative potential outside of the body under laboratory conditions and that the inflammatory phenotype demonstrated by the gene expression data did not affect cell growth rates in cHDPC cultures [5].

Our CFU data suggest the presence of dental pulp stem cells within our cultures. This cell type reportedly represents <1% of the total dental pulp cell population and provides an important population that can be activated to differentiate in vivo to enable tooth hard tissue repair by generating a mineralized extracellular matrix [29]. Despite the differences in CFU forming abilities, we subsequently demonstrated that both ncHDPC and cHDPC cultures were able to differentiate to give a mineralizing phenotype at similar levels when stimulated in vitro under appropriate exposure conditions. Clearly, the carious teeth included in this study still contained vital stem/progenitor cell populations, which implies that the regenerative capacity in the dental pulp was not significantly affected by the inflammatory and disease process [27]. Interestingly, the transcript level of SOX2 stem cell-associated transcript was also found to be significantly higher in cHDPCs than ncHDPCs (Figure 4).

Cells of the dentine-pulp complex include odontoblasts, fibroblasts, neurones, endothelial cells and stem cells, which can recognise invading bacteria due to their expression of a variety of pattern recognition receptors (PRRs), such as TLRs, nucleotide-binding oligomerization domain (NOD) receptors and nod-like receptors (NLRs) [30]. Among these receptor families, TLRs are considered to be the key participant in pathogen-associated molecular pattern (PAMP) detection. Notably, higher levels of TLR-2, -4 and -9 were detected in cHDPCs compared with ncHDPCs (Figure 4). These findings are in agreement with previous studies [31,32,33,34,35]; our data supports the importance of these receptors in the dentine-pulp complex’s immunological defence. TLR stimulation subsequently activates NF-κB and MAPK1 intracellular signalling pathways, resulting in the subsequent release of antimicrobial peptides, cytokines and chemokines, including IL-1α, IL-1β, TNFα, IL-4, IL-6 and IL-8 [16,36,37]. Notably, these cytokines, via autocrine signalling, can also activate these pathways and consequently their elevated levels may explain the upregulation of NF-κB and MAPK1 transcripts detected in the cHDPCs compared with the ncHDPCs (Figure 4). In addition, it has been demonstrated that these pathways play a dual role and not only escalate the pulp’s cytokine and chemokine inflammatory response [30,37], but they can also induce mineralisation events if local conditions are suitably permissive [38]. Consistent with this finding, we also detected a significantly increased expression of the dentinogenic-related transcripts of DMP1 and DSPP in cHDPCs compared with ncHDPCs (Figure 4). This finding was also in agreement with our ARS quantification data, wherein we found that there was a trend for higher mineralization levels in cHDPCs compared with ncHDPCs, although this difference was not significant. The elevated mineralising phenotype detected in cHDPCs may consequently relate to previous reports that indicate low-grade inflammation stimulates dentine-pulp complex repair events [16,39].

While our experimental results showed that transcript levels of cytokines were significantly higher in cHDPCs compared with ncHDPC cultures (Figure 4), which is consistent with previous studies [15,17,40,41,42], we are the first, to our knowledge, to demonstrate the maintenance of the activation of the IL-23/IL-17 signaling axis in cHDPC cultures. Retaining inflammatory signalling in culture will lead to autocrine feedback which will perpetuate this phenotype. Interestingly, others have also reported the presence of growth factors and chemokines produced by mesenchymal cells in conditioned media during cell culture. Our data may therefore indicate that cell derivation may significantly influence cell secretion, and this may have important implications for identifying which cells and culture conditions are optimal for use in secretome and exosome studies [43,44]. Currently, it remains unclear as to how this inflammatory phenotype is maintained in subsequent culture passages or whether further passage will cause levels to diminish. Studies which determine the presence of levels of bacterial components in culture passages, as well as epigenetic analyses, such as the determination of the methylation status of inflammation-associated gene promoters, may shed light on how inflammatory phenotypes are maintained in culture.

Furthermore, as our preliminary findings have demonstrated that relatively high levels of cytokines, including TNF-α, IL-6, IL-8, IL-17R and IL-23A, were expressed in cultures of cHDPCs. Consequently, it can be envisaged that immunotherapy might have application in chronic pulpal diseases [45]. As a next step, we are currently studying the effects of immunotherapy on inflamed dental pulp tissue in in vitro and in vivo models (unpublished data). We believe this research will identify new avenues in both experimental modelling and therapeutic treatments.

## 4. Methods and Materials

Ethical approval for this study was obtained from the University of Otago Ethics Committee (Health) (H20/13). Written informed consent was obtained from healthy patients who were non-smokers and had mature human permanent third molar teeth extracted for clinical reasons as part of a treatment plan at the Faculty of Dentistry Oral Surgery clinics, University of Otago, New Zealand. Prior to extraction, teeth were assessed for radiographic staging of carious lesions using the International Caries Classification and Management System (CMMS) [46] to ensure caries had not progressed to the inner third of dentine. Additionally, teeth were excluded from patients experiencing intense/spontaneous dental pain or dental pain disturbing sleep.

### 4.1. Cell Isolation and Culture

Freshly extracted non-carious/healthy and carious teeth (n = 5 each) were collected from patients who met the inclusion criteria as mentioned above.

Tooth surfaces were cleaned with sterile PBS (pH 7.5) and sectioned at 1 to 2 mm below the cementoenamel junction with the diamond cylinder bur fitted on a high-speed dental handpiece to access the pulp cavity. Coronal samples were placed in chilled Dulbecco-modified Eagle medium (DMEM) (Gibco^TM^, Grand Island, New York, NY, USA) and immediately transferred to the laboratory on ice for further processing. HDPCs were isolated using the standard explant approach and cultured following the methods previously described [47]. Briefly, pulp tissue excavated from the coronal pulp chamber of healthy/non-carious and carious teeth were dissected into pieces of ~1 mm^3^ and arranged in a single well of a 6 well plate containing DMEM (Gibco^TM^) supplemented with 20% FBS (Gibco^TM^), 1% antibiotic antimycotic (Gibco^TM^) and 0.5% gentamycin (Gibco^TM^) (henceforth referred to as cell culture media), and incubated at 37 °C with 5% CO_2_. Cell culture media were changed every third day in all experiments unless stated otherwise. Cultures were considered successful when emergence of HDPCs from the dental pulp explant was observed within a week and there was no evidence of contamination. Cultures were passaged when they were 70 to 80% confluent and experiments were conducted at passage 4.

### 4.2. Generation of Growth Curves

For cell growth analysis, ncHDPCs (n = 5) and cHDPCs (n = 5) were seeded at 19.2 × 10^3^ cells/well in 6 well plates (Thermo Fisher Scientific) and cultured for analysis at three time points. The plates were then cultured in an incubator at 37 °C with 5% CO_2_. Cell culture media were replaced every third day. At days 1, 3, 5 and 7, cells were detached by trypsinisation and stained with 0.4% trypan blue (Gibco^TM^) to enable viable cell counting using a hemacytometer (n = 5 per cell line) and corresponding growth curves were generated.

### 4.3. Colony Forming Capacity

A colony forming unit (CFU) assay [5] was performed by seeding 1000 cells in a 100 mm Petri dish (ncHDPCs (n = 5) and cHDPCs (n = 5)) and culturing under basal conditions, i.e., 37 °C with 5% CO_2_ for up to 14 days. The cell culture medium was changed every third day. At the end of the incubation period, the cell culture medium was discarded, the cells were washed twice with PBS and fixed by adding 4% paraformaldehyde (Gibco^TM^) (20 to 30 min) at room temperature. Subsequently, the cells were stained with 0.5% crystal violet solution (Merck, Darmstadt, Germany) for 30 min to visualize colony formation and were allowed to air dry prior to imaging (ChemiDoc^TM^ imaging system, BIO-RAD, Bio-Rad Laboratories Inc., Hercules, CA, USA). Cell colonies were quantified using plugins cell counter, Image J software (version:2.3.0/1.53f, Rasband, W.S., ImageJ, U. S. National Institutes of Health, Bethesda, MD, USA) in which >50 cells was classed as a single colony [48].

### 4.4. Immunocytochemistry

Cultured ncHDPCs (n = 5) and cHDPCs (n = 5) were seeded at 8 × 10^4^ cells/well into 8 chamber slides (Nunc^TM^ Lab-Tek^TM^ II chamber slides^TM^). After 48 h the culture medium was removed, and cells were fixed and permeabilized using 200 µL of fixation buffer (eBioscience^TM^, Life Technologies Corporation, Carlsbad, CA, USA) and 200 µL of permeability buffer (eBioscience^TM^), respectively, for 20 min. The slides were blocked using blocking buffer and then incubated with anti-vimentin antibody (Abcam, Abcam Inc., Cambridge, MA, USA) at a concentration of 5 µg/mL overnight at 4 °C. Normal rabbit Ig (Abcam) of at the same concentration as the primary antibody was used as a negative control. After overnight incubation, slides were washed and incubated with the secondary antibody (Dako, Hamburg, Germany) for 30 min followed by incubation with the 3,3′ diaminobenzidine (DAB, Thermo Scientific, Rockford, IL, USA) chromogen solution according to the manufacturer’s instructions (Dako REAL DAB + Chromogen). Counterstaining was performed using Gill’s hematoxylin, slides were overlayed with a coverslip and examined under light microscopy [47].

### 4.5. Differentiation Potential of Human Dental Pulp Cell Cultures

To assess the ability of HDPC cultures to differentiate down a mineralizing lineage, cultures were exposed to a mineralizing media (DMEM supplemented with 10% FBS + 5 mM β-glycerophosphate (Sigma-Aldrich, Saint Louis, MO, USA) + 100 µM ascorbic acid (Sigma-Aldrich) + 10 nM dexamethasone (Sigma-Aldrich). ncHDPCs (n = 5) and cHDPCs (n = 5) were seeded (19.2 × 10^3^ cells/well) in 6 well plates and when cultures became 70–80% confluent, the cell culture medium was replaced with the mineralizing media. Cultures were maintained in the mineralizing media for up to 21 days. Both ncHDPCs and cHDPCS cultures without exposure to the mineralizing media (DMEM containing 10% FBS only) served as controls. After 21 days, cultures were stained with Alizarin red stain (ARS) (ScienceCell^TM^ Research Laboratories, Carlsbad, CA, USA) to detect the presence of calcium deposits. Subsequently, any mineralized deposit produced by ncHDPCs and cHDPCs was quantified using an ARS staining quantification assay kit (ScienceCell^TM^ Research Laboratories) following the manufacturer’s instructions.

### 4.6. Quantitative Real-Time Reverse-Transcription Polymerase Chain Reaction

The total RNA was extracted from ncHDPCs and cHDPCs (n = 5 each) using a Purelink^TM^ RNA Mini Kit (Invitrogen^TM^, Carlsbad, CA, USA), and 300 ng RNA was reverse-transcribed to complementary deoxyribonucleic acid (cDNA) by using the Superscript^TM^ IV VILO^TM^ Master Mix with EZDNASE Kit (Invitrogen^TM^) according to the manufacturer’s protocol. qPCR was performed using TaqMan™ Fast Advanced Master Mix (Applied Biosystem™, Vilnius, Lithuania), gene-specific TaqMan™ Gene Expression Assays (Applied Biosystem™) and Quant Studio^TM^ 7 Flex Real-Time PCR System Machine (Applied Biosystem™). The following TaqMan gene expression assay primers were used: Toll-like receptor (TLR)-2, TLR-4, TLR-9, TNF-α, IL-1β, IL-6, IL-8, IL-17A, 1L-17R, IL-23A, dentine matrix acidic phosphoprotein (DMP)-1, dentine sialo phosphoprotein (DSPP), nuclear factor kappa B subunit 1 (NF-κB), mitogen-activated protein kinase (MAPK)-1, sex determining region Y-box (SOX)-2 and marker of proliferation Ki-67 (MKi67). Glyceraldehyde 3-phosphate dehydrogenase (GAPDH) and ARP1 actin-related protein 1 homolog (ACTR1)-B were used as reference genes (RGs). Primer information is provided in Table 1. Relative gene expression (2^−ΔCt^) was calculated using the cycle threshold (Ct) difference between HKGs and each transcript using the comparative Ct method. In brief, the Ct value obtained for each GOI and HKG was normalized to the mean of the RGs (GAPDH and ACTR1B). This was called the Δ Ct value (Δ Ct = Ct of GOI − Ct of RG).

The gene expression level (mRNA level) was expressed as 2^−ΔCt^ and was calculated using Microsoft Excel (Microsoft, version 16.54, Redmond, Washington, DC, USA). The fold change (2^−ΔΔCt^) for each gene was calculated using the equation:2^−ΔΔCt^ = 2^−ΔCt(cHDPCs)^/2^−ΔCt(ncHDPCs)^

(The 2^−^^ΔCt^ used above was the average gene expression value for each group, ncHDPCs and cHDPCs).

### 4.7. Statistical Analysis

A statistical analysis was performed using GraphPad Prism version 9.0.2, GraphPad Software (La Jolla, CA, USA). The data were evaluated for normality using the Shapiro-Wilk test. If the data were normally distributed, an unpaired student’s *t*-test was used, otherwise a Mann–Whitney test was performed. For comparisons, a value of *p* < 0.05 was considered to indicate a significant difference.

## 5. Conclusions

This study has shown that cHDPCs retain their stem cell potency and inflammatory phenotype following culture. Our studies also demonstrate that the inflamed tooth pulp contains viable stem cell populations which have the potential for expansion, proliferation and differentiation to enable soft as well as hard tissue repair. These findings are important as they can be used to inform the use of REPs in diseased teeth and therefore further studies are warranted which identify novel therapeutic approaches to harness the regenerative potential of stems cells from diseased teeth.

## Figures and Tables

**Figure 1 ijms-23-09425-f001:**
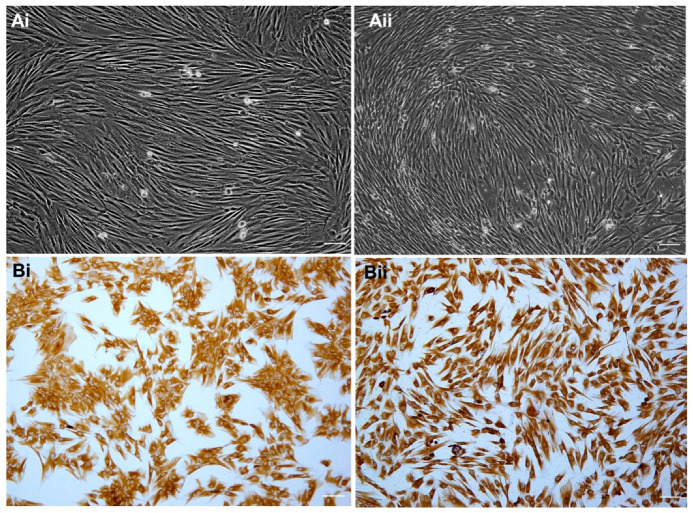
Representative photomicrographs showing: (**A**) confluent cultures of (**i**) spindle-shaped ncHDPCs and (**ii**) cHDPCs, and (**B**) vimentin positive (**i**) ncHDPCs and (**ii**) cHDPCs. All cultures were at passage 4. Scale bar = 100 µm.

**Figure 3 ijms-23-09425-f003:**
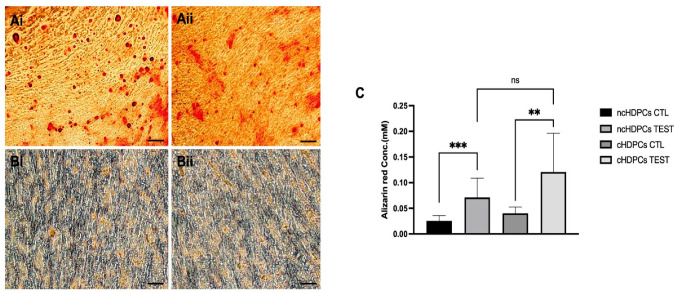
Representative photomicrographs of ncHDPC (**Ai**) and cHDPC (**Aii**) cultures after Alizarin red S-staining showing in red the presence of calcium deposits within the background of differentiated dental pulp cells on day 21 of culture in mineralizing media. Cultures of ncHDPCs (**Bi**) and cHDPCs (**Bii**) in standard unsupplemented culture media, which served as controls. Scale bar = 200 µm. (**C**) Quantification of Alizarin red S-stain in cultures of ncHDPCs and cHDPCs (n = 5 each) (TEST and CTL/CONTROL represent HDPCs cultured with or without mineralizing media for up to 21 days, respectively). A significant difference was found between ncHDPC test and control cultures (*** *p* = 0.0004), and there was a similarly outcome for cHDPC test and control cultures (** *p* = 0.0048). No significant difference (ns) was found between mineralization levels in ncHDPC and cHDPC test cultures (*p* = 0.18). Error bars indicate mean ± SD.

**Figure 4 ijms-23-09425-f004:**
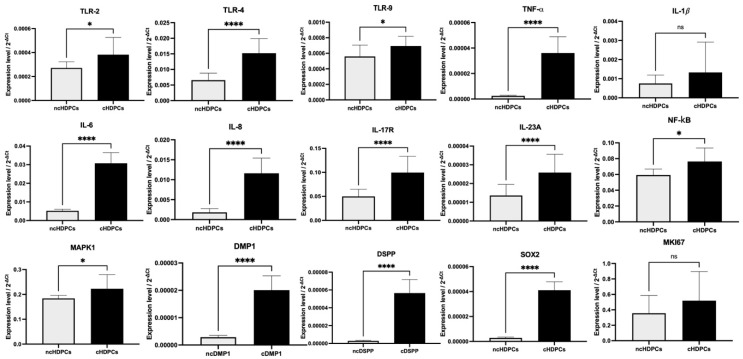
Gene expression assay by quantitative real-time PCR analysis for markers relating to inflammation (TLR2, TLR4, TLR9, TNF-α, IL-6, IL-8, IL-17R, IL-23A, NF-κB, MAPK1), dentinogenesis (DMP1, DSPP), pluripotency (SOX2) and proliferation (Ki67) in cultures at passage 4 (n = 5 each). Values are presented after normalization to average of GAPDH and ACTR1B levels. * Indicates * *p* < 0.05, **** *p* <0.0001, ns = non-significant. Error bars indicate mean ± SD.

**Table 1 ijms-23-09425-t001:** Details of genes studied.

	Gene Symbols	Gene Name	Gene Alias	Assay ID
Dentinogenic markers	DMP1	Dentine matrix acidic phosphoprotein 1	ARHP	Hs01009392_m1
	DSPP	Dentine sialo phosphoprotein	DFNA39	Hs00171962_m1
Inflammatory markers	TLR-2	Toll like receptor 2	TIL4	Hs00610101_m1
	TLR-4	Toll like receptor 4	ARMD10	Hs00152939_m1
	TLR-9	Toll like receptor 9	CD289	Hs00370913_s1
	IL1-β	Interleukin 1 beta	IL1F2	Hs01555410_m1
	TNF-α	Tumour necrosis factor	TNFSF2	Hs00174128_m1
	IL-6	Interleukin 6	BSF-2	Hs00174131_m1
	IL-8	C-X-C motif chemokine ligand 8	IL8	Hs00174103_m1
	IL-17RA	Interleukin 17 receptor	IL17R	Hs01056316_m1
	IL-17A	Interleukin 17A	IL17	Hs00174383_m1
	IL-23A	Interleukin 23 subunit alpha	IL-23	Hs00372324_m1
Transcription markers	NFκB1	Nuclear factor kappa B subunit 1	NF-kappa-B	Hs00765730_m1
	MAPK1	Mitogen-activated protein kinase 1	ERK, p38,	Hs01046830_m1
Proliferative marker	MKi67	Marker of proliferation Ki-67	MIB-1	Hs00606991_m1
Stemness marker	SOX2	Sex determining region Y-box 2	ANOP3	Hs04234836_s1
Reference genes	GAPDH	Glyceraldehyde 3-phosphate dehydrogenase	G3PD	Hs02786624_g1
	ACTR1B	ARP1 actin-related protein 1 homolog B	ARP1B	Hs00194899_m1

## Data Availability

The data presented in this study are available on request from the corresponding author.

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
