# Peer review of "Potentiality and Inflammatory Marker Expression Are Maintained in Dental Pulp Cell Cultures from Carious Teeth"

_ijms, 2022, doi:10.3390/ijms23169425_

Round 1

Reviewer 1 Report

I would like to compliment the authors for their scientifically sound investigation, the rigorous data reporting and the overall content of their manuscript which is of great importance for the endodontic field and of great interest for the endodontic readership.

The study aims to shed some more light on an important topic, that is the characteristics of dental pulp cells derived from carious teeth. This topic has basic research ramifications, but more importantly bears significant clinical translation. A dental pulp affected by caries should not be deemed invariably and a priori severely compromised, with accumulating evidence constantly demonstrating the reparative capacity of the tissue. However, cariously affected teeth may exhibit lower survival rate, especially due to restorative concerns as a result of the extensive decay. This means that, not that seldom, dental pulp tissues with cellular content already primed for repair may need to be extirpated. Therefore, the present study, addresses and provides basic research evidence for two clinically relevant scenarios, namely, cases in which cariously affected pulps may be left in situ after proper treatment is administered and cases where diseased pulps (considered either salvageable or unsalvageable based on a tentative pre-op and a definitive intra-op diagnosis) are extirpated and destined for cell isolation with the purpose of subsequent cell transplantation therapies in pulpless immature or mature teeth (revitalization or "regenerative endodontic procedures"). 

General remarks:

Information that is partly missing and would be relevant for the readers to know is some clinical description of the caries status of the affected teeth that were extracted (besides the radiographic staging), and, maybe even more importantly, the pulpal status, as this could be assessed pre- and intra-operatively. In particular, what has been the pulpal diagnosis of the cariously affected third molars extracted? Can we extrapolate the findings of the present study to all diseased pulp tissues encountered in the clinical practice? Or should we be selective in the cariously diseased pulps that we should consider as tissues with good healing capacity or tissues capable of providing dental cell populations with good regenerative potential? This is information that could be added in the manuscript in the Methodology section. In addition, an idea for future research could be the comparative evaluation and correlation between clinical pulpal/histological diagnosis and phenotypic characterization of the cells isolated. Lastly, the exposure of the carious dental pulp cells to dentin matrix proteins isolated from the carious-affected teeth would be another idea for future research as supplementation of culture media with DMPs would simulate better the environment that these cells encounter during caries insult.

Comments on Introduction section:     

Introduction is well-written, funneling from general information to more specific, while being concise. It reads smoothly and aims and objectives are clearly presented at the end of the section. Maybe some information from relevant investigation addressing the same topic would be useful (that is to say, besides only stating that there is a paucity of of evidence on the specific topic, maybe briefly highlighting some points from similar studies; however, this is covered later on in the Discussion section and therefore is a point of marginal concern).

Comments on Methods section:

Clear presentation and sound methodology. In sub-section 4.5, I assume that the information missing is 100 μM ascorbic acid and 10 nM dexamethasone.

It is nice that the authors have conducted such a comprehensive gene expression analysis, ranging from TLR receptors to pro- and anti-inflammatory genes and specific transcription factors association with inflammatory cell phenotypic changes. For cell proliferation, maybe Ki-67 immunocytochemical staining (instead of investigating at the gene expression level) would be even better. Minor remark: use the term "reference gene" instead of "housekeeping gene". Nice to see that the gene expression results were expressed with the 2-ΔΔCt method, the most appropriate and informative method.

Comments on Results section:

Results are again very clearly and succinctly presented. Figures are of high quality and sufficiently supported by the text provided. Maybe sub-section 2.2 should be named "Colony forming capacity", followed by the "Differentiation potential of human dental pulp cell cultures" sub-section.

An interesting finding is the significant upregulation of the dentinogenic genes in the cHDPCs which is not equally reflected to the amount of the extracellular calcium deposits compared to the ncHDPCs (however there is a clear trend observed).

Comments on Discussion section:

The point raised in "General remarks" (see above) regarding the clinical description of the pulpal status and the conditions encountered upon pulp tissue retrieval is also relevant here when discussing the differences between the present study and similar ones. For instance, the fact that no problems were found in isolating dental pulp cells in the present study may be also related to the better histological condition of the puplp tissue samples obtained (besides the larger tissue volume that the pulp tissue occupies in third molars).

In general, all results are critically discussed in relation to existing evidence while plausible hypotheses (e.g., the autocrine positive feedback loop between cytokines and transcription factors) are also provided. Furthermore, ideas for follow-up studies, based on the evidence provided by the present study are mentioned.

Comments on Conclusions section:

No special remarks, conclusions are well supported by the evidence provided earlier.

Reviewer 2 Report

Introduction

62. The statement “There is only limited literature describing the isolation and characterization of HDCPs derived from carious teeth, the cellular and molecular differences between ncHDPCSs and cHDPCSs” needs references.

Methods. 

229-236. The authors reported that cells were isolated from human pulp tissue of normal and infected teeth obtained from healthy and no-smoker patients. However, they have not considered the demographic data of the patients included in their study.

232-234. The statement “Prior to extraction, teeth were assessed for the radiographic staging of carious lesions using the International Caries Classification and Management System” turns out to be lacking because there is no mention of the clinical and radiographic criteria used to identify the carious lesions.

Author Response

This manuscript is a resubmission of an earlier submission. The following is a list of the peer review reports and author responses from that submission.